# Lived experience in mental health research in Ghana and Indonesia: What have we learned?

Hannan Legend Tizaa[1], Lisa Agyinor Forson Aboagye[2], Esenam Abra Drah[3], Joseph Stanley Ofosuware[4], Elisa Faustina[5], Desty Endah Nurmalasari[5], Inda Marlina[5], Sarah Khairunnisa Budiyanto[6], Agus Sugianto[7], Lily Kpobi[8], Annabella Osei-Tutu[8], Diana Setiyawati[9], Wulan Nur Jatmika[9], Erminia Colucci[9,10], Ursula Mary Read[11]*

1 BasicNeeds Ghana, Tamale, Ghana, 2 Independent researcher and mental health advocate, Agona Swedru, Ghana, 3 Independent researcher and mental health advocate, Osiem, Ghana, 4 Independent researcher and mental health advocate, Accra, Ghana, 5 Independent researcher, Yogyakarta, Indonesia, 6 Universitas Jakarta Internasional, Jakarta, Indonesia, 7 University of Manchester, Manchester, United Kingdom, 8 University of Ghana, Accra, Ghana, 9 Universitas Gadjah Mada, Yogyakarta, Indonesia, 10 Middlesex University London, London, United Kingdom, 11 University of Essex, Colchester, United Kingdom

* ursula.read@essex.ac.uk

## Abstract

There are increasing calls for the involvement of people with lived experience in mental health research. However, to date there are few examples of peer research conducted by people with lived experience of mental health conditions from the Global South. This paper explores the experiences of peer researchers involved in mental health research in Ghana and Indonesia. Peer researchers with lived experience of mental health conditions were employed as part of the research team to carry out qualitative and participatory arts-based research. Following this, peer researchers completed feedback forms and written reflections as well as taking part in unstructured discussions on their experience. Together with the academic research team, themes were developed from this feedback to identify the benefits, challenges and lessons learned from this process. Peer researchers benefited from developing skills and confidence, sharing lived experience, opportunities to engage with stakeholders and a supportive working environment. However, they identified several challenges including balancing care for self and others, precarious working conditions, enduring power imbalances and limited training and preparation. Key lessons included the need to consider safety and support needs, preparation for working with participants with lived experience, meeting resource needs and the importance of involving peer researchers across the research cycle. Based on these experiences, we identify several recommendations for peer research, particularly in Global South settings. These include involving people with lived experience in research design and costing, careful preparation and training, creating safe spaces and enabling access to mental health support, providing fair and comprehensive remuneration, creating opportunities for career development and democratizing opportunities for participation.

**Data availability statement:** We do not report on formal research data in this paper. The content reported is derived from personal verbal and written reflections and discussions between peer researchers with lived experience of mental illness and academics at the University of Ghana, Universitas Gadjah Mada, Indonesia, the University of Middlesex and the University of Essex (formally the University of Warwick and King's College London). The methods are intersubjective and cannot be replicated. The content of these informal discussions is not suitable for sharing and secondary analysis due to the personal, sensitive, subjective and context specific nature of these reflections. Ethical approval for the first project was provided by King's College London SSHL Research Ethics Subcommittee (HR/DP-20/21-22551), Universitas Gadjah Mada Faculty of Psychology Research Ethics Committee (REC) (HR/DP-20/21-22551) and the University of Ghana Ethics Committee for the Humanities (ECH) (ECH 129/ 20-21). For the second project ethical approval was provided by Universitas Gadjah Mada Faculty of Psychology REC (8856/UN1/FPSi.1.3/SD/PT.01.04/2023) and the University of Ghana ECH (ECH 223/22-23) and the University of Warwick Biomedical and Scientific Research Ethics Committee (BSREC 131/22-23). Ethical approval in the UK was then transferred to the University of Essex (ETH2324-2045) following a change of institution by the Principal Investigator. A data management plan was developed for both studies regarding the governance and sharing of research data collected. This data is not reported in this paper. For queries around access to data for the second study researchers should contact: Universitas Gadjah Mada Faculty of Psychology Research Ethics Committee Email: fpsi@ugm.ac.id, The University of Ghana Ethics Committee for Humanities Email: ech@ug.edu.gh , The University of Essex REO Research Governance Team Email: reo-governance@essex.ac.uk For the first study researchers should contact: Universitas Gadjah Mada Faculty of Psychology Research Ethics Committee Email: fpsi@ugm.ac.id , The University of Ghana Ethics Committee for Humanities Email: ech@ug.edu.gh , King's College London Research Governance Office Email: rgo@kcl.ac.uk

## Author summary

This paper highlights the experiences of people with lived and living experience of mental health conditions who are engaged in research (also called peer researchers) in two countries in the Global South. We worked with peer researchers as part of research teams conducting qualitative and arts-based research on mental health in Ghana and Indonesia. Through feedback forms, reflections, and discussions, peer researchers shared their experiences of the process. The benefits for peer researchers included gaining research skills, building confidence, sharing personal stories, and working in a supportive environment. However, peer researchers also faced challenges, such as balancing personal well-being with research demands, dealing with limited pay, and navigating power imbalances. The paper recommends several steps to improve peer research, especially in the Global South. These include involving people with lived experience in planning and budgeting for research, providing adequate support, creating safe research environments, offering fair pay, and creating opportunities for career growth. It also stresses the importance of providing access to mental health care and ensuring the research process is inclusive and empowering for all involved.

## Introduction

Until recently, there has been limited involvement of people with lived experience in mental health research in the Global South [1,2]. A review of Patient and Public Involvement (PPI) in low-and-middle-income countries (LMICs) found that this primarily consisted of advisory boards made up of community stakeholders, rather than people with lived experience conducting research themselves [3]. In peer research, by contrast, people who are the focus of the research are involved in carrying it out [4]. Peer researchers (or lived experience researchers) are recognized as 'experts by experience' due to their first-hand knowledge of living with a mental health condition. They draw on this experience to conduct research and interpret findings [5].

Including people with lived experience as mental health researchers is supported by arguments for disability rights and inclusion as articulated in the slogan 'nothing about us without us' [6]. In the Global South this also aligns with moves to decolonise global mental health research and value diverse forms of knowledge and understanding [7]. Indeed, people with lived experience in the Global South may experience multiple intersecting forms of exclusion and discrimination based on race and socio-economic status as well as their mental health [8,9]. Entrenched inequalities arising from colonial histories within Global North-South research partnerships can also impact on peer researchers who are likely to be more vulnerable to extractive research practices. Advocates with lived experience from the Global South, including the Global Mental Health Peer Network (GMHPN), have argued for greater inclusion of people with lived experience in research on equal terms [10] and major research funders now require their involvement [2,11].

**Funding:** The research for this paper was supported by the following awards: UKRI Arts and Humanities Research Council International Networks for Disability Inclusive Development [Grant number: AH/X009637/1] Developing a network for mutual learning on the potential of creative arts for mental health advocacy and activism in Ghana and Indonesia UR, EC, AOT, LK, DS; UKRI Global Challenges Research Fund Arts and Humanities Research Council [Grant number: AH/V013548/1] The impact of COVID-19 on people with psychosocial disabilities in rural and urban settings in Ghana and Indonesia and priorities for inclusive recovery UR, EC, AOT, LK, DS; British Academy International Writing Workshops [Grant number: WW22\100194] Qualitative and Visual Mental Health Research in Ghana and Indonesia EC, UR, LK, DS; University of Warwick Economic and Social Research Council Impact Acceleration Account [Grant number: ES/T502054/1] Using film to engage stakeholders in Ghana and identify priority actions for mental health UR, AOT, LK; University of Warwick Arts and Humanities Impact Fund [Grant number: AHIF.1.OC.004] Using film to engage stakeholders in Ghana and identify priority actions for mental health UR, AOT, LK, THL. The funders had no role in the study design, data collection and analysis, decision to publish, or preparation of the manuscript.

**Competing interests:** Lily Kpobi is on the editorial board for PLOS Mental Health and had no role in the review of this paper and decisions regarding publication.

## Benefits of lived experience in mental health research

Aside from these arguments for inclusion and representation, peer research has been claimed to bring benefits to peer researchers themselves, as well as to research processes and outcomes. Peer researchers can benefit from personal development and increased confidence, as well as from opportunities to learn new skills and share their expertise [12,13]. People with lived experience offer unique knowledge and insight which can promote reflexivity and dialogue, challenge assumptions and enrich research findings [2,10]. They also have first-hand experience of stigma and discrimination [14], as well as the particular challenges of accessing treatment and support in a low-income context. Shared experiences between peer researchers and research participants can help to build trust and rebalance power [4]. Peer researchers can bring empathy and understanding based on their own experiences of mental distress and help-seeking. In the Global South this can include experiences of using formal and informal services, such as traditional and faith healers. This can help in recruiting participants, particularly those who are more marginalised, and facilitate greater openness and sensitivity, both in conducting research and in reporting research findings. Finally peer research can be particularly impactful in engaging stakeholders with lived experience perspectives. This can help to humanise research findings and challenge stigma [14]. Peer research can provide opportunities to build networks and to use research findings for advocacy.

## Challenges for lived experience research in mental health

Despite these potential benefits, there are several challenges. Peer researchers can feel under pressure to disclose their lived experience without consideration of the potential distress and discrimination which may arise. Because of the novelty of peer research in Global South settings, people may fear increased exposure to stigma and discrimination if they disclose their mental health status. Symptoms of mental illness and discrimination can impact on education and work opportunities, meaning that people living with mental health conditions can be left behind compared to their peers. Consequently, people with lived experience may fear that they do not have the necessary skills and training to effectively participate in research. Additionally, peer researchers are often not accorded the same level of respect as their academic counterparts, particularly where academics are new to the concept of peer research. Experiential knowledge is commonly undervalued compared to the theoretical and scientific knowledge that dominates academic discourse [5,8,15]. As a result, the unique perspectives that peer researchers bring to research are frequently overlooked or underappreciated, limiting the potential of peer research to provide transformative insights and challenge the status quo. With increased requirements to involve people with lived experience in research there is a danger that this becomes a 'check-box exercise', with the risk of perpetuating rather than dismantling inequalities [16].

Furthermore, disregard for economic rights and principles of transparency, mutuality and equality remain commonplace in lived experience research [10]. Peer

researchers are often employed on short-term contracts and there are few substantive research positions, particularly in the Global South [10,17]. Accommodations and support may not be in place to enable people living with mental health conditions to work equitably and to their best ability [4]. Though these are concerns globally, this is particularly important to consider in the Global South where it can be harder to access support and there can be fewer avenues for redress in the event of exploitative and discriminatory workplace practices. Providing support and accommodations can also be challenging within the strictures of funding applications which call for detailed advance breakdown of costs, with limited flexibility and generally tight deadlines.

## Peer research in Ghana and Indonesia

While there has been an increase in mental health advocacy by people with lived experience in Ghana and Indonesia, to date there has been very little direct involvement of people with lived experience as researchers, as in much of the Global South [1]. However the inclusion of lived experience researchers is supported by several recent developments. In both countries there has been increasing advocacy to promote the rights of people living with mental health conditions, including their civic and political inclusion. In both countries advocacy groups have been established led by people with lived experience such as Champions for Mental Health International in Ghana and Komunitas Peduli Skizofrenia Indonesia (KPSI). In Ghana people with lived experience have also been involved in advocacy and anti-stigma campaigns [18,19]. These initiatives have encouraged some people to speak publicly about their mental health struggles in community spaces such as places of worship, in local and international media and online. Finally, the expansion of global mental health research in Ghana and Indonesia, including projects involving the co-authors [19–22], has created opportunities for the involvement of people with lived experience, for example as participants in advisory boards [19], in co-designing interventions [19] and in research and engagement activities using arts-based and participatory methods [20,21,23]. This has helped to increase their visibility and representation in mental health research.

This paper reflects on the experiences of peer researchers engaged in mental health projects in Ghana and Indonesia. In this paper we use the terms 'peer researchers' and 'lived experience researchers' interchangeably to refer to people with first-hand experience of mental health conditions who were employed as part of the research team to conduct research activities. Although some of the peer researchers also have academic roles, in this paper we use the term 'academic researchers' to refer to researchers employed at academic institutions who do not identify as lived experience experts. We describe the successes and difficulties encountered in peer research and the facilitators and barriers to meaningful involvement of people with lived experience in research in these settings. The aim of this paper is to illustrate the lessons learned from this experience and provide signposts to guide lived experience and academic researchers engaging in mental health research, particularly those working in the Global South.

## Methods

### Ethics statement

Ethical approval for the first project was provided by King's College London SSHL Research Ethics Subcommittee (HR/DP-20/21–22551), Universitas Gadjah Mada Faculty of Psychology Research Ethics Committee (REC) (HR/DP-20/21–22551) and the University of Ghana Ethics Committee for the Humanities (ECH) (ECH 129/ 20–21). For the second project ethical approval was provided by the University of Warwick Biomedical and Scientific Research Ethics Committee (BSREC 131/22–23), Universitas Gadjah Mada Faculty of Psychology REC (8856/UN1/FPSi.1.3/SD/PT.01.04/2023) and the University of Ghana ECH (ECH 223/22–23).

For both projects, all participants in interviews, participatory video workshops and lived experience groups provided written informed consent, however this paper does not report on these findings. Whilst peer researchers are named co-authors of this paper, we use ID numbers rather than names for verbatim quotes to enable an open discussion of both positive and negative experiences.

## Recruitment of peer researchers

The nine peer researchers who co-authored this paper were employed as research assistants on two research projects funded by United Kingdom Research and Innovation (UKRI) which used participatory, qualitative and arts-based methods. The first in 2021–2022 explored the impact of COVID-19 on people living with serious mental illness in Ghana and Indonesia. The second in 2023–2024 supported groups of people with lived experience and artists in Ghana and Indonesia to develop a network to explore the potential of creative arts for mental health advocacy and activism. A total of 12 peer researchers were employed on the two projects: four peer researchers from each country were recruited for the first project and two peer researchers in Ghana and four in Indonesia were recruited for the second. The six academic researchers, who are also co-authors, and Ghanaian peer researchers ED and OJ worked on both projects.

For both projects lived experience researchers were recruited via word of mouth through existing connections within service providers, NGOs and advocacy groups, as well as advertisements on social media and mental health-related WhatsApp groups. Applicants were interviewed online or face-to-face by members of the academic research teams.

## Training for peer researchers

Throughout the research process mentoring and guidance were provided by academic researchers at the University of Ghana (Accra) and Universitas Gadjah Mada (Yogyakarta) with support from the UK research team via monthly and ad hoc online meetings. For the first project, peer researchers were trained by the academic researchers in research methods, including interview skills, participatory video and thematic analysis, and in ethics and safeguarding through online and face-to-face workshops. Peer researchers were also trained in responding to participant distress using an adapted distress protocol [24]. Peer researchers completed short versions of the WHO Recovery Planning template [25] to describe signs of relapse, triggers and accommodation and support needs. They also identified people they wished to be contacted in the event they needed support. This information was shared with the academic researchers in each country. The lead academics in Ghana and Indonesia are trained psychologists with clinical as well as research experience. They held debriefing sessions with the peer researchers during research activities and if needed directed them to professional services.

## Peer researcher activities

For the first project the peer researchers in each country were engaged in research planning, sampling and recruitment of participants, taking consent and conducting interviews alongside academic researchers. Following initial training, the peer researchers co-facilitated a participatory action research workshop with caregivers and people with lived experience to review and refine the interview topic guides. Interviews with people living with mental health conditions, caregivers and stakeholders were conducted in Accra and Tamale in Ghana, and Jakarta and Yogyakarta in Indonesia. Peer researchers were trained in thematic analysis [26] and worked with the academic research teams to develop visual maps of key themes and sub-themes from the interviews. Following this, the research team, including the peer researchers, co-facilitated a three-day participatory video workshop in Tamale, Ghana and Yogyakarta, Indonesia. The former was held in person with the Ghana and UK researchers, the latter was held in person with the Indonesia research team but with remote participation by the UK facilitators due to COVID-19 restrictions. During the workshops, peer researchers and participatory film-makers/researchers supported research participants with lived experience and caregivers to develop short films illustrating their experiences during the COVID-19 pandemic.

For the second project peer researchers assisted with recruiting and engaging people to join a lived experience group. Some were recruited from advocacy organisations of which the peer researchers were part. Peer researchers worked with academic researchers and artists to plan and facilitate group meetings and participatory arts activities. In Indonesia the peer researchers for this project were also artists and so drew on these skills as well as their lived experience. Arts activities were chosen by the groups and included drama, song, dance, poetry and painting.

For both projects, peer researchers are actively involved in disseminating research findings, including presenting at online and in-person national and international film-screenings, seminars and conferences. With additional impact funding, participatory videos from the first project were shown to participants and stakeholders in Tamale, Ghana and peer researchers co-facilitated a discussion on actions to promote social inclusion for people living with mental health conditions. In Indonesia, peer researchers helped to plan and facilitate a performance event for stakeholders and members of the public. This featured drama, song, painting and poetry developed by the group based on their lived experience. The performance was followed by a meeting with stakeholders, such as local government and representatives from health and social services to share some of the arts-based narratives and engage in a discussion on policy and support implications. These events were attended by a peer researcher from Ghana (ED) who supported the Indonesian peer researchers with a reflection exercise as part of the evaluation (Table 1).

## Reflection and evaluation

As part of research reporting and evaluation, peer researchers completed a short feedback form as well as unstructured written reflections on their experience. The idea for this paper was developed during a British Academy international writing workshop led by EC, UR, DS and LK. Peer researchers from both countries took part in monthly online webinars over two years where they further reflected on their experiences in a dedicated break-out group. Some also periodically met online without the academic research team.

Although 12 peer researchers were employed across the two projects, in this paper we reflect as co-authors on the experiences and perspectives of 9 peer researchers - 4 from Ghana and 5 from Indonesia. The Ghana peer researchers developed the first draft of the paper with the academic research team during an in-person writing workshop in Ghana in November 2023. Together we reviewed the written reflections from Ghana and discussed peer researchers' experiences. We grouped together key themes which we used to structure this paper. The draft paper was then shared with Indonesian

**Table 1. Peer researcher activities.**

| | Project 1 | Project 2 |
|---|---|---|
| **Project aim** | To understand the impact of COVID-19 on people with psychosocial disabilities in Ghana and Indonesia and inform guidelines for inclusive recovery | To develop mutual learning on using creative arts to facilitate dialogue, advocacy and activism to promote inclusion and participation of people with psychosocial disabilities in Ghana and Indonesia |
| **Number of peer researchers** | 4 Ghana<br>4 Indonesia | 2 Ghana (also employed on project 1)<br>4 Indonesia |
| **Research methods** | • Participatory workshop to develop topic guides<br>• Qualitative interviews<br>• Participatory video workshops<br>• Film screenings and stakeholder discussions (Ghana) | • Participatory groups using arts-based activities<br>• Arts-based performance event and stakeholder meeting (Indonesia) |
| **Peer researcher activities** | • Attending online and in person planning meetings<br>• Co-facilitating participatory workshop<br>• Recruiting participants<br>• Taking consent<br>• Conducting interviews<br>• Analysing findings<br>• Co-facilitating participatory video workshops<br>• Planning and facilitating film screenings and stakeholder discussions (Ghana only)<br>• Process evaluation and reflection<br>• Disseminating research findings (film screenings, presentations, publications etc.) | • Attending online and in person planning meetings<br>• Recruiting participants<br>• Taking consent<br>• Planning and organising participatory group meetings<br>• Co-facilitating participatory group meetings and creative arts activities<br>• Planning and organising an arts-based performance event in Indonesia<br>• Planning and facilitating a stakeholder meeting in Indonesia<br>• Process evaluation and reflection<br>• Disseminating research findings (film screenings, presentations, publications etc.) |

peer researchers. Following their review the themes were expanded and revised and new themes added based on their written reflections. Academic researchers also reviewed the paper and added additional reflections based on their experience of the research process. We continued to refine and revise the themes during online discussions. We use verbatim quotes from the written feedback forms and reflections as well as some verbatim extracts from conversations between the peer research teams to convey the direct experience of peer researchers. Written reflections and inter-country discussions were in English. Verbal discussions within the Indonesian team were in Indonesian. Where needed, members of the Indonesian research team translated their verbal reflections from Indonesian to English.

### Research team characteristics

Lived experience researchers: Peer researchers from Ghana are Ghanaian nationals based in the Northern, Central, Eastern and Greater Accra regions. Peer researchers from Indonesia are Indonesian nationals based in Java. Though they represent some diversity in terms of ethnicity and socio-economic background, all but one are university graduates and all have competence in English. This represents a level of advantage compared to the average. All are involved to varying degrees in local mental health advocacy groups and some in international advocacy activities, for example with the GMHPN. Only four (THL, SB, IM, AS) had prior experience of conducting academic research in mental health. Beyond lived experience and advocacy work, peer researchers also brought varied training and skills relevant to the research process, for example in IT, communication skills, community engagement and art.

Academic researchers: The academic research team includes two researchers from Ghana (AOT, LK), three from Indonesia (DS, WNJ, AS), one White Southern Italian (EC) and one White British (UR), both employed at UK universities. All the academic researchers also have training and experience as mental health practitioners (LK, AOT, DS, WNJ, EC psychology, UR occupational therapy). Longstanding relationships between some of the peer and academic researchers, as well as power relations discussed below, may have inhibited an open discussion of some of the challenges. However, peer researchers within each country also spent time reflecting on their experiences alone and with other members of the peer research team. The use of English for discussions and written outputs across the international team disadvantaged Indonesian team members and inhibited a more fluent and nuanced discussion.

## Findings

### Motivations and expectations

Peer researchers were motivated to take part in the projects through a concern to create change, develop knowledge and skills and engage with others with similar experiences:

> 'I saw it as an opportunity to collaborate using my lived experience and the lived experiences of others to make a difference in creating awareness and influencing policy change.' (PR2, Ghana)

> 'To participate in this project would be a great opportunity for me to serve for an issue I care about. I also wished to deepen my knowledge and exchange insight with the community.' (PR6, Indonesia)

> 'I was recently diagnosed with bipolar disorder, and at that time, I was eager to learn more about mental health from both a personal and academic perspective. Becoming a peer researcher seemed like a great way to connect with others who were going through similar things, and to turn my experiences into something that could help others.' (PR8, Indonesia)

> 'I have been interested in mental health issues for years. For all that time I put myself as an activist. When I was offered to be a peer researcher I said to myself that it was a good chance for me to broaden my knowledge and enrich my experience.' (PR5, Indonesia)

As described in the final quote, peer research built on similar motivations for working as mental health activists and shared many of the same aims. For those new to research there was some concern about the demands of the peer researcher role and the particular expertise, time and commitment needed. For those with prior research experience outside the sphere of mental health, the peer researcher role brought the topic closer to personal experience and posed the question 'How different will this role be?' One peer researcher expressed anxiety about the kind of questions she was required to discuss with participants around a very sensitive topic:

'At first, I found it quite difficult to approach the questions. I was afraid my questions would be offending informants and caregivers.' (PR7, Indonesia)

Peer researchers involved in advocacy for some time were confident about sharing their lived experience, however for others this was new. Some expressed concerns about whether they would be taken seriously if people knew about their mental health difficulties:

'I was worried a lot about the right thing to do. It was like whether [it] is all right to tell the participants that I am a person with [a] psychosocial disability. I was worried that the participants would see me as someone that is incapable of being a peer researcher.' (PR6, Indonesia)

Though lived experience was explicitly recognised and valued, peer researchers appreciated that they were not obliged to always disclose their mental health condition:

'I was particularly enthused by encouragement from the academic researchers that I don't have to reveal my lived experience status under any circumstances I don't feel like doing it.'(PR3, Ghana)

### Benefits of working as peer researchers

We worked together to group peer researcher reflections on what was valued in the role into four themes: developing skills and confidence, sharing lived experience, opportunities to engage with stakeholders and a supportive working environment.

### Developing skills and confidence

Despite their apprehensions several peer researchers found that the training and approach built up their skills and confidence:

'I expected the research to be challenging for a person with no research background or experience. The role was made easier through the training and participatory research approach'. (PR1, Ghana)

'I didn't know I could easily do this. Public speaking and interacting with others were a challenge for me. But I believe because of the training we were given interaction was much easier. Now I can stand anywhere and talk without any fear or shivering'. (PR2, Ghana)

Having knowledge and opinions as peer researchers welcomed and given equal consideration was an important aspect of this process, particularly given that peer researchers had experienced their views being dismissed in the past. This confidence was evidenced during participatory thematic analysis of interviews for the first project when we engaged in lively debates between peer and academic researchers.

Peer researchers also increased their knowledge and understanding around mental health and gained confidence in balancing the demands of work with managing their mental health:

*'I have to learn a lot about mental health not only for people but also for myself. I have to balance between working hard and keeping my mental condition stable. It is not easy but the benefit for me is really worth it.' (PR6, Indonesia)*

Some peer researchers valued their increased understanding of the research process, including aspects such as research funding. They also gained insight into the messy practicalities of research, for example the ways in which research plans and timelines could be disrupted:

*'The most important thing I have learned as a peer researcher was to be flexible in handling events. A lot of things might change during the process of creating the events, so we have to be flexible to fit in.' (PR6, Indonesia)*

**Sharing lived experience**

Peer researchers highly valued the opportunity to share their lived experience in an accepting environment with others who had experienced mental health issues:

*'I liked the opportunity to exchange experiences with persons with disability in an environment that was free from stigmatization and discrimination as well as non-judgemental.' (PR3, Ghana)*

Peer researchers spoke about how they were able to draw on their lived experience to empathise with participants, even when they came from quite different backgrounds. This could facilitate the data collection process and encourage people with lived experience to take part in research:

*'I learned that everyone's unique experience and perspective is valuable. I also realized that being able to relate my own experience helped me become more empathetic while conducting interviews, allowing me to connect with the participants.' (PR8, Indonesia).*

*'During the data collection, I met this lady who was shy and hesitant to speak, so I told her that I am also a service user whose conditions have improved because of my willingness to open to others. So, if you also speak out, help could come, and your situation would also improve like mine. After this we had a hearty and lengthy conversation, which made her happier.' (PR3, Ghana)*

Another peer researcher described how she was also able to use her experience as a 'survivor' to encourage and inspire others:

*"Those in my group were inspired by the time I shared about being productive whilst surviving, on how to still be ourselves, be accepting about it […] I was happy I could be a good influence when they opened their shell. I feel like a proud parent and hope for their growth to better life in the future.' (PR5, Indonesia)*

Peer researchers in both projects were involved in community engagement and recruitment. Peer researchers who had existing relationships with the community through their advocacy work could help ensure cultural sensitivity and develop trust:

*'A 'we' sense was developed by those involved in the research, because of the established relationship of peer researchers with participants and relevant stakeholders. This consequently made the research process seem a collective responsibility'. (PR3, Ghana)*

Peer researchers in both settings were also inspired through developing connections with and learning from the experiences of leading mental health advocates with lived experience within the research teams. For example, during a meeting DN described how she was inspired and encouraged by working with AS who has gained international recognition for his advocacy work.

## Opportunities to engage with stakeholders

Aside from creating shared connections with people with lived experience, peer researchers in both countries also valued how the role gave them the opportunity to interact with, and potentially influence, key stakeholders in their country, including those in government agencies.

*'In everyday life, it is rare to have an interaction with the CHRAJ [Commission on Human Rights and Administrative Justice] or social welfare boss to seek information from them. Assuming this role gave me the opportunity.' (PR4, Ghana)*

Engaging with high-status stakeholders was also seen to contribute to gaining recognition and respect:

*'My involvement in mental health research has brought me into contact with very important people in society which has increased my self-esteem and self-worth. And because my family and friends see my pictures with these personalities on social media, I have gained more acceptance and respect among them as a result.' (PR3, Ghana)*

## A supportive working environment

Peer researchers in Ghana reflected on the 'team spirit' within the research team which they felt created a friendly and supportive working environment. Some peer researchers formed a close bond which extended outside of the formal work environment. Peer researchers discussed the value of mutual support and encouragement, particularly when someone was feeling stressed or down.

*'After each day's work, I could not just wait for the next day because I believed it was equally going to be a whole new experience and an exciting one. We were there for each other on the field, so I had no fears at all.' (PR4, Ghana)*

Some peer researchers had previously experienced discrimination in the workplace. They therefore valued working in a team where they could be open about their lived experience whilst at the same time being recognised as a team member with important skills, opinions and responsibilities. For example, one peer researcher explained how they valued 'the ability to openly share our views as peer researchers and how it was welcomed and encouraged at every point' (PR4, Ghana)

Peer researchers appreciated that, whilst support needs were recognised, they were also trusted to work independently and assume responsibility for tasks such as preparing and planning arts-based workshops. Peer researchers were actively involved in scheduling research tasks around their particular needs and preferences. A peer researcher described how she felt that the impact of her mental health condition on her productivity was acknowledged for the first time:

*'Growing up, people would not take my words and wouldn't believe me how my holistic health condition is. I was often thought [of] as lazy when in reality I work the most to my limit. In no other place would this limitation be acknowledged and given understanding.' (PR5, Indonesia)*

## Challenges of working as peer researchers

Despite these perceived benefits, peer researchers also discussed several challenges. These were not openly discussed by all the peer researchers and reported experiences differed between the country teams, based in part on their varied roles and responsibilities. Academic researchers also expanded on the challenges based on their own reflections and

experience, particularly related to research processes. We grouped these challenges into four themes: balancing care for self and others, precarious working conditions, enduring power imbalances and limited training and preparation.

## Balancing care for self and others

One of the biggest challenges for peer researchers was managing their own mental health whilst also providing emotional support and reassurance to participants. This was particularly a concern during participatory research groups for the second project where participants met several times and shared challenges from their lived experience. Although, as described above, some peer researchers learned to balance these demands, others identified a need for more support and safeguarding measures. Some peer researchers described how the work took a high toll on their emotional resources leading to feelings of 'burn out'. This was particularly the case when working closely in participatory workshops which were felt by Indonesian peer researchers to be emotionally 'intense':

*'I needed to be a caregiver for every participant and other fellow survivors as they're fragile. Any slightest mistake I do in my gesture, face expression, choice of word, tone, timing and more could be a potential trigger for them.' (PR5, Indonesia)*

One peer researcher described struggling to communicate effectively with participants with mental health difficulties and how this could trigger her own distress:

*'Their condition made me suffer a lot on how to communicate effectively. They're tense. They avoid eye contact. They're either very self-centred or have no self-esteem at all. There is no in-between. And this triggered me often as well but I need to get the job done. I only work half day but the fatigue lasted a few days. This was hard.' (PR5, Indonesia)*

Another peer researcher described her difficulties in navigating the roles of 'researcher' and 'peer'. This could be particularly tricky for those who were also academic researchers and taking on a 'lived experience' identity:

*'I found it challenging to navigate my positioning—whether to identify more as a researcher or as a 'peer.' Since I was newly diagnosed and had only been in treatment for less than a year, I sometimes felt that I didn't have enough 'lived experience' to fully relate. Additionally, some conversations with participants who shared similar experiences were unexpectedly triggering for me.' (PR8, Indonesia)*

Peer researchers in Ghana also discussed 'compassion fatigue'. Although academic researchers provided support during participatory group meetings, members of the groups could contact peer researchers independently or through WhatsApp groups. Due to the difficulties in accessing affordable mental health care in Ghana, this could place an additional burden on the academic and peer researchers to provide support. In addition, academic researchers reflected that participants often have high expectations of research interventions, including support to access medication and establish livelihoods. Disappointed expectations carried additional weight for peer researchers who have first-hand experience of these difficulties This meant that they could feel especially bad when they were unable to meet participants' expectations.

## Precarious working conditions

Another challenge for peer researchers was that they were employed on casual and short-term contracts. Academic researchers observed that due to project-based research, irregularities in funding and limited opportunities, peer researchers are often left without substantive employment. Most had no other sources of income and no regular salary and

none were in receipt of any form of welfare benefits. This precarious situation was worsened by inflation and significant increases in the cost of living in both countries during the research. One peer researcher stated that in their view:

*'The payment was just adequate but could be improved in the future in view of the recent hikes in cost of living.' (PR3, Ghana)*

Other peer researchers felt that their pay should have reflected different levels of responsibility and experience:

*'Peer researchers in this project have different capacities. Payment should be based on previous research experience' (PR7, Indonesia)*

*'I had double job desks in this project for quite some time, I did some all nights, rented assistants, yet I only got paid as a peer researcher. I feel exploited at some points when other facilitators and peer researchers did not do as much as what I have done' (PR5, Indonesia)*

In addition to this, academic researchers experienced long delays transferring funds from institutions in the UK to universities in Ghana and Indonesia. While academic researchers received monthly salaries from their university roles, peer researchers on casual contracts could be unpaid until funds arrived. Whilst the academic researchers worked hard to find ways around this (borrowing from other funds, pre-financing research activities from their own pockets), this only underlined peer researchers' precarity and deepened existing inequities between resources available to academic and peer researchers (and between UK and Ghana/Indonesia researchers).

*'I was quite surprised when informed that the payment would come late […] Luckily UGM provided early substitution for us. But it's really just awkward and impractical to receive money then to transfer it back later. It's like I'm in debt.' (PR5, Indonesia)*

## Enduring power imbalances

Closely related to the above were persisting power imbalances between academic and peer researchers related to wide differences in status and access to resources. Though academic researchers made efforts to bridge these, peer researchers could struggle to challenge actions, attitudes and structural barriers which perpetuated discrimination, disempowerment or disadvantage. Some peer researchers felt that they were not provided with opportunities to take on more responsibilities. Assumptions, anxieties and protectiveness may have prevented academic researchers from stepping back to enable lived experience researchers to take more active roles:

*'At first I thought that I will be involved in designing the material for the seminars, creating the activities, deciding the themes and handle everything. However, I feel that I only helped a little in terms of handling the participants in seminars.' (PR6, Indonesia)*

## Limited training and preparation

Some peer researchers felt that they were not adequately prepared for the research and had insufficient training and capacity building. For the first project this arose in part from restrictions imposed by the pandemic. However, academic researchers reflected that short project timelines, limited funding and institutional delays also impacted on the time dedicated to training and preparation and placed peer researchers under pressure to complete activities within a short timescale. The second project, which was concerned with participatory advocacy rather than primary data collection, did not include structured training.

Consequently, some peer researchers felt unprepared and confused about the aims of the research. This made it difficult for them to explain the research to participants:

*'Peer researchers must understand about the research so they can explain more clearly to participant when there is question […] When the participants asked me about things related to the research it was hard for me to explain' (PR6, Indonesia)*

Academic researchers felt that the limitations of both projects' scope, funding and timelines impacted on capacity-building for peer researchers to acquire skills to participate equitably at all stages of the research process, for example carrying out data analysis, presenting findings and writing reports. Some peer researchers expressed a desire for more training to address this:

*'I expected a lot of capacity building opportunities through the project. These expectations were met in a way but not fully since the number of trainings were not many.' (PR3, Ghana)*

Peer researchers also found online training difficult to engage with and preferred in-person and practice-based sessions where they could ask questions and seek clarification:

*'The training was very thorough and prepared me well for conducting interviews and anticipating interactions with participants. However, since it was conducted online, I still felt somewhat unprepared for the realities of face-to-face situations.' (PR8, Indonesia)*

Peer researchers had not previously met in-person and face-to-face meetings also enabled them to connect and build relationships for mutual support and learning.

## Lessons learned

Based on the above, we identified several key lessons including the need to consider safety and support needs, preparation for working with participants with lived experience, training and resource needs, the importance of peer researchers being involved across the entire research cycle and the value of participatory arts-based methods.

## Safety and support needs

Peer researchers stressed the importance of considering their safety and support needs throughout the research process. Prior to taking up the role, peer researchers were anxious about how best to manage their mental health while conducting research. For example, some were concerned about what would happen if they experienced a relapse. Peer researchers were also concerned about how their mental health condition might impact on their performance: 'how my condition gets in the way, especially in tasks involving a lot of brain work' (PR1, Ghana).

Lived experience researchers who had been out of work for some time due to discrimination and difficulties finding employment, or who worked irregular hours, could struggle to adjust to a working routine. Those taking medication could experience drowsiness, particularly in the mornings. Providing structure, support and jointly agreed working practices, as well as flexibility to adjust schedules and timelines when people needed rest or time out, was important to provide an inclusive working environment:

*'Having been unemployed, I was used to waking up irregularly but to be properly involved in the projects I had to adjust my daily routines with the help of my phone alarm and colleagues. Initially I was sluggish and unsure in my dealings with research participants and stakeholders, but with encouragements from senior researchers and adherence to the guidelines, I eventually fit in.' (PR2, Ghana)*

Consideration was also given to aspects such as travel time and childcare (one peer researcher and co-author is a mother of young children). Peer researchers on the first project valued regular breaks between research tasks, flexible working hours, not being 'bombarded with instructions' and not too many tight deadlines.

*'Filling out the [recovery planning tool] gave me assurance right from the start that I was in good hands. And on the field, I got to have some breaks that refreshed me to start another interaction. The breaks and early closing were very helpful.' (PR4, Ghana)*

Given the lack of research experience, peer researchers valued being members of a team and that they were not expected to conduct research alone. Instead they worked alongside academic researchers who provided supervision and guidance. A supportive team environment also created a space where ideas could be freely expressed:

*'I think the most important support for a peer researcher with lived experience is the support system or the people who work with them. A good support system will make it easier for peer researchers to express ideas because there will be less pressure.' (PR4, Ghana)*

Peer researchers valued being able to easily access mental health support during the research process from experienced academic researchers:

*'During the participatory workshop at the University of Ghana, there was a day I had an issue at home and this, in addition to workshop stress, drove me close to relapse. A senior researcher cleverly noticed it and talked to me as well as gave me a place to cool off before rejoining the rest of the team.' (PR1, Ghana)*

## Preparing to care for others

Although peer researchers had themselves experienced mental health crises, academic researchers reflected that they should not assume that this prepared them to respond confidently and effectively when research participants became distressed. As described above, this could be triggering for peer researchers, expose them to 'secondary trauma' or leave them feeling overwhelmed by the responsibilities of caring for their own mental health and, at the same time, responding to the needs of others.

For the first project, training and protocols were provided by the academic researchers on communication skills and how to respond if participants became distressed. One peer researcher explained how this helped him to overcome his anxiety around his dual status as a researcher and a person living with a mental health condition:

*'From the start, I found it difficult to interact with the respondents because I found myself struggling to deal with 'status-ambivalence' but when I had the guidance and shared experience from senior researchers, it made it easier.' (PR3, Ghana)*

Others described how they gained skills through practice:

*'I have to make sure that every participant feels safe. I learned to take care of myself and take care of all participants.' (PR6, Indonesia)*

However some peer researchers who were involved in co-facilitating participatory groups with people with lived experience felt that they needed additional training to increase their knowledge and skills so as to care effectively for others and for themselves:

*'Prior training on how to be a proper caregiver for the participants and how to survive as someone with mental health problems while tending for others.' (PR4, Ghana)*

*'The additional training that I think I would need is psychological first aid (PFA) training. When we were talking to caregivers and informants, sometimes they share traumatizing experiences. PFA is also important to check on peer researcher condition.' (PR7, Indonesia).*

Peer researchers also reflected that while they might share experiences of mental health conditions with participants, there were important differences, such as symptoms experienced and the length of time living with the condition.

### Resource needs

Peer researchers were paid monthly salaries as well as travel and transport costs and per diem allowances during fieldwork. However, as described above, delays in payment and lack of access to institutional support placed peer researchers at a serious disadvantage. Internet access could also prevent equitable participation. In Ghana there is limited free Wi-Fi, and internet data must be purchased on demand. Though the research budget included costs for internet access, delays meant that at times peer researchers could not afford to attend online meetings. This placed responsibilities on the research team to take pre-emptive action to ensure everyone could participate equitably.

Peer researchers were paid in line with local university base pay scales for research assistants. However, this did not account for additional costs arising from their mental health condition. In both Ghana and Indonesia psychiatric medication and psychological interventions must often be paid for out of pocket. A peer researcher in Indonesia explained how she planned to use her salary to offset the costs of medication:

*'That amount of money may not be so much even in the Rupiah value, but as someone living with debilitating chronic illness, that amount of money could help ease my medication bills.'*

*(PR5, Indonesia)*

Although travel insurance had been costed for internal and international travel for the peer researchers as this was not covered by the research institutions, there was no budget for health insurance. The peer researchers suggested that health insurance costs should be included in the salary to meet treatment costs and ensure they were working to their best ability:

*'The cost for BPJS [government health insurance] as long as the project goes would be nice. As medication and therapy are the sole maintenance for mental illness survivors. Good medication access will be able to keep the workflow nice and stable.' (PR5, Indonesia)*

Finally two peer researchers also noted that peer research involved 'emotional labour/cost' which should also be recognised in calculating fair compensation for their work:

*'I believe that peer researchers should be compensated in a way that recognizes both the time and emotional labour involved' (PR8, Indonesia)*

As a consequence of this emotional labour, one peer researcher suggested that the cost of psychological treatment should also be included.

### Involving people with lived experience across the research cycle

Academic researchers reflected that though organisations of people with lived experience were involved as partners, individuals with lived experience were not directly engaged in the design process for the first project. Whilst three peer researchers (AS, OJ, ED) were involved in developing the second project, other peer researchers were recruited only after funding had been obtained. This led to some confusion around expectations and limited understanding of research aims. Some peer researchers in Indonesia felt training should clarify the research process and approach, the research schedule, expectations and deliverables. They highlighted the importance of involving peer researchers across the research process, not just in data collection:

> *'I was hoping to be more involved in the research planning, data analysis, and report writing, not just in the data collection process. While I enjoyed gathering data and interacting with participants, I was a bit surprised that my role didn't extend further into the other research stages' (PR8, Indonesia)*

Peer researchers identified a need for further training in areas such as data analysis, communication skills and public speaking, leadership, organising events, and mental health advocacy to enable them to take a more active role across the research cycle.

Whilst some peer researchers were involved in data analysis and developing research outputs and dissemination activities, academic researchers noted that competing commitments meant that this could be inconsistent, particularly as time elapsed and peer researchers were no longer paid from project funds. In Ghana this was mitigated to some extent by the re-employment of two peer researchers on the second project. In addition, academic researchers noted the need to step back and create opportunities for peer researchers to take the lead when they were ready to do so.

### The value of participatory arts-based methods

Both peer and academic researchers noted that the use of participatory arts-based methods, including film, storytelling, painting, music, poetry and drama, proved valuable in sharing lived experience whether between peer resarchers, with research participants, or with stakeholders and the general public. Peer researchers assisted artists in facilitating arts-based activities during the participatory groups as well as taking part in these activities themselves. While peer researchers perceived some challenges in their use, such as different levels of skill and confidence in using creative techniques, we found arts-based methods, such as participatory video, provided a safe, inclusive and accessible way of engaging with others and discussing sensitive issues:

> *'I found creative arts to be an effective and natural outlet for expressing our views on mental health issues without excessive exposure, offering us a means to communicate emotions and experiences that may be challenging to convey through traditional methods.' (PR1 Ghana)*

> *'Utilizing creative and arts-based methods as part of inclusive approaches is an effective strategy to ensure broad participation […] Creative art serves as a powerful medium for mental health advocacy, as it is an effective tool for challenging stigma and communicating complex messages in an accessible and impactful way.' (PR9 Indonesia)*

## Discussion

This paper reports on our experiences pioneering peer research in mental health in Indonesia and Ghana. Given the discrimination and exclusion experienced by peer researchers within families, universities, workplaces and in society at large, being able to work as part of a team where lived experience was valued was a powerful form of recognition. However, as we have shown, the process was not without challenges arising from structural hierarchies, misunderstandings and

missteps. Acknowledging and discussing these openly is the first step towards enabling meaningful and equitable involvement [9]. Whilst some of the issues we highlight, such as equity, inclusion and safeguarding, are applicable globally, others were exacerbated in these settings due to colonial histories, which favour Global North institutions and the English language. We faced difficulties in managing funds between Global North and South institutions, which exacerbated socio-economic inequalities within and between members of the research team. Furthermore, while lived experience research is comparatively well-established in the UK and US with some leading academics in the field, peer research is in its infancy in the Global South. Our experience also highlights the importance of careful preparation and planning, to enable truly inclusive ways of working.

Claudia Sartor [10] outlines nine key principles for lived experience involvement in decision making which equally apply to research. These include mutual respect and trust, transparency, non-discrimination, non-tokenism, reasonable accommodation, flexibility, diversity and equality, empowerment and clear communication. Our experiences underscore the value of these principles, whilst highlighting the ways in which, even with the best intentions, research teams can fall short of adhering to them in practice. Our recommendations for involving people with lived experience in research in the Global South, align with those made by Sartor and others, including the recent WHO framework on meaningful engagement [27].

## Recommendations

**Involve people with lived experience in research design.** People with lived experience should be involved from the inception of grant applications to shape the research questions and research design. They should also be involved in budgeting decisions to ensure they are not disadvantaged and that all their costs are accounted for. This is particularly important to consider when working in Global North-South partnerships when decisions can be taken under pressure to meet deadlines without taking time and care to involve all partners equitably. Since most lived experience researchers are not based in academic institutions this also requires funding streams and capacity development to support people with lived experience in the Global South to develop and ultimately lead research proposals.

## Create safe spaces and provide mental health support

It is vital to create safe and inclusive spaces to support peer researchers' wellbeing. This includes intentionally making space to discuss difficult feelings that may arise during research, including triggering events or conversations. Debrief and reflection sessions should be built into the research design and consistently and carefully facilitated. Art-based methods and creative activities can help in developing safe spaces for reflection and facilitating communication around difficult topics [28]. It is also important to provide access to mental health support when needed, whether from professional providers, charitable and advocacy organisations or other sources. This is particularly an issue to consider when working in Global South settings where mental health services can be difficult to access and may incur significant upfront costs.

## Provide fair and comprehensive remuneration

It is essential to value the time and expertise of peer researchers by providing equitable remuneration [10]. In the Global South this includes not only costing for salaries consistent with their level of expertise, but also including funds to meet 'hidden costs' such as internet access, insurance and treatment expenses. This could involve a 'disability premium' on salaries or budgeted items in research costing. This is particularly important when peer researchers are on short-term or consultancy contracts. This may require greater flexibility on the part of research funders, for example in moving funds across budget lines or developing new budget lines. Funders and budget administrators can develop ways to enable research teams to respond promptly to unforeseen events, whilst ensuring ethical practice and accountability. Remuneration should also reflect differences in skills and responsibilities within peer researcher as within academic research teams, recognising that peer researchers not only bring their lived experience of mental health conditions to research, but other relevant skills, knowledge and experience. It is also important to account for the vulnerability of researchers in the Global South to national

and global economic upheavals. During the course of the research both Ghana and Indonesia experienced very high inflation which directly impacted on research costs as well as daily living expenses, for example higher internet, food and transport costs. Ultimately there is a need to create more substantive posts for lived experience researchers in the Global South, including within research and advocacy institutions, to enable them to access institutional supports and benefits.

## Provide sufficient training and preparation

Peer researchers need training and capacity-building opportunities to develop skills and competencies to participate in research on an equitable basis and ultimately take on responsibility and leadership, rather than be relegated to simpler, administrative tasks [29] or data collection alone. This requires ensuring training and capacity building is high quality, accessible and responsive to individual learning needs and abilities. Training should include skills required across the research process, from grant writing to disseminating findings, including writing and presentation skills. Careful preparation for research includes clarity on expectations and responsibilities of all team members, including peer researchers. These should be negotiated in a transparent and inclusive way at the outset of the project rather than imposed in a top-down manner by academic researchers. Capacity building should be provided not only for peer researchers but also for academic researchers to develop skills to work meaningfully, ethically and effectively with peer researchers in a respectful and empowering way. This training should be led by and co-produced with experts with lived experience and include decolonising ways of working to address intersectional inequalities across international research teams.

In addition, it should not be assumed that peer researchers have the skills needed to work with participants with mental health difficulties based simply on their lived experience. Training should include providing mental health support and responding to issues which might arise in the course of research, whilst ensuring that peer researchers can easily link participants to professional services where needed.

Though online training is widely promoted as cost-effective and convenient, in our experience this was inferior to face-to-face learning which enabled the building of trusting relationships and provided more flexibility in tailoring learning support to differing needs. Online training can also reinforce inequalities where internet access is costly and unreliable.

## Ensure accessibility and flexibility

Our experiences underscore the importance of providing safe, accessible and flexible working environments and reasonable accommodations so that peer researchers can contribute to research on an equal basis. Examples include longer timelines, adjusting work schedules, and building in time for rest and reflection. Peer researchers may need to take a break to attend to their mental health. This requires forward planning by research teams as well as flexibility and responsiveness from funders and research institutions. For example, there may be the need to provide additional training and funding as well as adjustments to project timelines to cover the costs of a temporary replacement should a peer researcher be unable to work. This also points to the need for changes to the competitive and pressurised cultures within academic institutions to create more inclusive ways of working.

## Create opportunities for growth and career development

Just as academic researchers have career goals, for people with lived experience working in research can be a significant step in their career path. Opportunities for employment as lived experience experts are growing in the Global South but remain constrained compared to the Global North. It is important to consider from the outset how the project could support peer researchers towards achieving their career goals. This will vary. Some may want to develop a career in research or study for further qualifications, as with some of the authors of this paper. Others may wish to enhance their advocacy skills or develop other transferable skills. Possibilities to support peer researchers in identifying next steps and meeting their goals include mentoring and building in opportunities and costing for further training [30].

## Democratize opportunities for participation

To date, some groups are under-represented in peer research and there is a tendency to treat people with lived experience as a homogenous group [9]. As we have described, there were differences in the peer researcher group in terms of lived and living experience of mental health conditions, as well as other life experiences and professional backgrounds. It is important to consider these intersectional differences when recruiting peer researchers, such as diversity in ethnicity, age, gender and socio-economic status so as to include a plurality of perspectives from within as well as across countries [17]. This means creating opportunities for more people with lived experience to be involved in research, including those from more disadvantaged backgrounds and minoritised groups, such as those with little or no formal education and who do not speak English, those living in remote areas and people from older age groups, as well as people with varying experiences of and perspectives on mental health conditions. Barriers can be reduced by diversifying approaches to recruitment, intentionally recruiting lived experience researchers from under-represented groups, providing accessible training and support and developing inclusive methodologies, such as participatory arts-based methods, where different forms of knowledge and experience can be heard and valued. There is potential in developing new spaces for lived experience research beyond academia [17], for example, research could be led by lived experience advocacy groups, which are growing in number in the Global South, or arts-based organisations. Consistent with a social model of disability, mental health conditions should not in themselves be a barrier to participation. Rather, the onus is on research teams, institutions and funders to provide support and accommodations so that even people with profound disabilities can have opportunities to meaningfully participate.

## Conclusion

While the involvement of people with lived experience in mental health research is increasingly valued, this is still a new phenomenon in the Global South [1]. In this paper we discussed our learning from two research projects employing lived experience researchers in Ghana and Indonesia. Our experience highlights the value of peer research and its potential to counter discrimination, engage communities, enrich research findings and extend research impact. We identify several areas where careful planning, training and budgeting could create a more equitable experience for peer researchers and empower them to achieve their full potential. We also highlight where academic researchers, research funders and institutions could make changes to support peer research in mental health. However, we are mindful that entrenched power structures within institutions and wider society require systemic changes which can be much more challenging to overcome [9,29,31]. Working with peer researchers in settings such as Ghana and Indonesia requires additional attention to structural and intersectional inequities both between differently positioned members of Global South research teams as well as within North-South partnerships [32]. As in all participatory methods, there is real risk that peer research can reinforce inequalities, where those who are most disadvantaged bear the greatest cost and remain marginalised within decision-making [33].

## Acknowledgments

We gratefully acknowledge the support of the facilitators and members of the British Academy International Writing Workshop on Qualitative and Visual Mental Health Research in Ghana and Indonesia where we worked together to develop this paper for publication. We are grateful to Grace Ryan and colleagues from the UPSIDES study who shared training and safeguarding materials which were developed or adapted for these projects [34]. Above all we express our thanks to the research participants and members of the lived experience groups in Ghana and Indonesia.

## Author contributions

**Conceptualization:** Hannan Legend Tizaa, Lisa Forson, Esenam Drah, Joseph Ofosuware, Agus Sugianto, Lily Kpobi, Annabella Osei-Tutu, Diana Setiyawati, Erminia Colucci, Ursula Mary Read.

**Data curation:** Lily Kpobi, Annabella Osei-Tutu, Diana Setiyawati, Wulan Nur Jatmika, Erminia Colucci, Ursula Mary Read.

**Formal analysis:** Lisa Forson, Esenam Drah, Joseph Ofosuware, Inda Marlina, Sarah Khairunnisa Budiyanto, Lily Kpobi, Annabella Osei-Tutu, Diana Setiyawati, Wulan Nur Jatmika, Erminia Colucci, Ursula Mary Read.

**Funding acquisition:** Agus Sugianto, Lily Kpobi, Annabella Osei-Tutu, Diana Setiyawati, Erminia Colucci, Ursula Mary Read.

**Investigation:** Hannan Legend Tizaa, Lisa Forson, Esenam Drah, Joseph Ofosuware, Elisa Faustina, Desty Endah Nurmalasari, Inda Marlina, Sarah Khairunnisa Budiyanto, Agus Sugianto, Lily Kpobi, Annabella Osei-Tutu, Diana Setiyawati, Wulan Nur Jatmika, Erminia Colucci, Ursula Mary Read.

**Methodology:** Agus Sugianto, Lily Kpobi, Annabella Osei-Tutu, Diana Setiyawati, Wulan Nur Jatmika, Erminia Colucci, Ursula Mary Read.

**Project administration:** Agus Sugianto, Lily Kpobi, Annabella Osei-Tutu, Diana Setiyawati, Wulan Nur Jatmika, Ursula Mary Read.

**Supervision:** Lily Kpobi, Annabella Osei-Tutu, Diana Setiyawati, Erminia Colucci, Ursula Mary Read.

**Writing – original draft:** Hannan Legend Tizaa, Lisa Forson, Esenam Drah, Joseph Ofosuware, Agus Sugianto, Lily Kpobi, Annabella Osei-Tutu, Diana Setiyawati, Erminia Colucci, Ursula Mary Read.

**Writing – review & editing:** Hannan Legend Tizaa, Lisa Forson, Esenam Drah, Joseph Ofosuware, Elisa Faustina, Desty Endah Nurmalasari, Inda Marlina, Sarah Khairunnisa Budiyanto, Agus Sugianto, Lily Kpobi, Annabella Osei-Tutu, Diana Setiyawati, Wulan Nur Jatmika, Erminia Colucci, Ursula Mary Read.

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
