## [Decision Letter · Decision Letter 0]

PMEN-D-24-00484

Lived experience in mental health research in Ghana and Indonesia: What have we learned?

PLOS Mental Health

Dear Dr. Read,

Thank you for submitting your manuscript to PLOS Mental Health. After careful consideration, we feel that it has merit but does not fully meet PLOS Mental Health’s publication criteria as it currently stands. Therefore, we invite you to submit a revised version of the manuscript that addresses the points raised during the review process.

We look forward to receiving your revised manuscript.

Kind regards,

Kaaren R Mathias, PhD

Academic Editor

PLOS Mental Health

Journal Requirements:

1. Please provide an Author Summary. This should appear in your manuscript between the Abstract (if applicable) and the Introduction, and should be 150–200 words long. The aim should be to make your findings accessible to a wide audience that includes both scientists and non-scientists. Sample summaries can be found on our website under Submission Guidelines:

https://journals.plos.org/globalpublichealth/s/submission-guidelines#loc-parts-of-a-submission.

Additional Editor Comments (if provided):

Thanks for this excellent paper submitted. Please make edits to respond to reviewers.

Reviewers' comments:

Reviewer's Responses to Questions

**Comments to the Author**

1. Does this manuscript meet PLOS Mental Health’s publication criteria ? Is the manuscript technically sound, and do the data support the conclusions? The manuscript must describe methodologically and ethically rigorous research with conclusions that are appropriately drawn based on the data presented.

Reviewer #1: Yes

Reviewer #2: Yes

Reviewer #3: Yes

Reviewer #4: Yes

Reviewer #5: Yes

Reviewer #6: Yes

2. Has the statistical analysis been performed appropriately and rigorously?

Reviewer #1: N/A

Reviewer #2: N/A

Reviewer #3: N/A

Reviewer #4: Yes

Reviewer #5: Yes

Reviewer #6: N/A

3. Have the authors made all data underlying the findings in their manuscript fully available (please refer to the Data Availability Statement at the start of the manuscript PDF file)?

Reviewer #1: No

Reviewer #2: Yes

Reviewer #3: No

Reviewer #4: Yes

Reviewer #5: Yes

Reviewer #6: No

4. Is the manuscript presented in an intelligible fashion and written in standard English?

Reviewer #1: Yes

Reviewer #2: Yes

Reviewer #3: Yes

Reviewer #4: Yes

Reviewer #5: Yes

Reviewer #6: Yes

5. Review Comments to the Author

Reviewer #1: This is an interesting paper with thorough information, addressing various aspects. However, I believe it is too verbose, making it challenging to hold the reader's attention. It would be better to revise it before considering it for publication.

Reviewer #2: Overall this was an excellent exploration of the incorporation of those with lived experience into the research process especially as it pertains to mental health. I appreciated the plain language and practical considerations that were discussed within the paper. I saw the biggest strengths of the work to be the clarity of scope in terms of who exactly was included as peer researchers and the straight forward nature in which the difficulties of such an approach were described. The data used enriched my understanding of the research and humanized the research experience in a way that was apropos given the topic.

I appreciated the recommendations section's (starting on on line 723) categorization of the various needs when considering including lived experience within a research team, however, I felt that the section lacked enough specificity to be fully operationalized by researchers seeking to follow the suggestions.

While the article is focused on the concept of including those with lived experience within mental health research the arts based nature of the interventions is mentioned but not described in enough detail to fully understand what was done exactly. This may not be relevant given the focus of this particular article, but I thought it might have been helpful to more broadly understand what was asked of the peer researchers.

In some sections of the article the "Global South" is discussed as potentially a unique environment in terms of challenges to implementation of lived experience inclusive research in mental health. However, I did not get enough information on why this may be the case nor did I get a sense as to whether the recommendations and discussion is valuable globally or more appropriate to this particular region.

I very much appreciated the acknowledgement that is made in the research team characteristics section (starting on line 220) that the diversity in terms of educational and socio-economic background was limited. I had hoped to see more discussion regarding the potential challenges of broad inclusion in mental health research of those with lived experience due to symptoms and challenges of the lived experience itself. The challenges section (starting on line 71) and in several other places in the paper seem to concentrate mostly on power systems not allowing participation, but I think that a more nuanced conversation would be helpful when considering that direct symptoms of mental health could be highly challenging to experience while engaging in the research process. Especially when attempting to research more severe mental health challenges, inclusion of those with lived experience in the design and implementation of the research is not just a hurdle of willingness and funding. This is briefly touched on in the section entitled democratize opportunities for participation, however, this section mostly indicates that it should be done without offering concrete solutions on how to do it.

I very much enjoyed the article and am glad that this is being pushed more firmly into our consciousness as researchers in mental health. Thank you for your work and your insights. I hope that you are able to continue down this path and continue to do good scholarship that helps those that have been overlooked for far too long.

Reviewer #3: Thank you for the opportunity to review this important article. The manuscript is well written and organized overall and very interesting findings that fit with meaningful inclusion of patients, public and people with lived experience of mental illness. This paper adds an important contribution to the literature in advancing PPI or Patient Oriented Research and addresses a large gap with inclusion of people with mental health conditions as peers in conducting research.

While the paper advances knowledge in this area, particularly form the Global South perspective, it is unclear how the thematic analysis was conducted? for example themes are described well with excerpts /examples but what kind of thematic analysis was conducted? e.g. Bruan and Clarke reflexive thematic analysis? this might help add rigor to the methods section.

One line 223 authors note " all peer researchers are graduates and 5 have master's degrees" it is unclear what graduates means? graduates of what? perhaps consider taking this out and simply saying 5 had graduate degrees?

This adds more nuanced understanding of peer researchers as not only having lived experience of mental health conditions but also other experiences in which they can meaningfully contribute to research e.g. some may be engineers, lawyers, activists etc. See DOI:10.1186/s40900-021-00266-1

Finally, I appreciate the spotlight on low middle income countries and learned a lot about how people with mental illness are treated in the Global South. I find it interesting that many of the themes are relevant to the "Global North" and Global contexts. This may warrant policy directions related to thinking interjectionally about race, class, gender, geography etc. for policy on PPI in mental health research. A statement to this effect may strengthen the conclusions and overall findings and recommendations.

Overall, this manuscript and work is very informative,

Thank you for the opportunity to review the important work.

Reviewer #4: The abstract was well presented and summarized clearly explained what was done and the teams involved .It actually provided a clearer direction into the manuscript. However I felt there was need for the number of researchers to be mentioned in the abstract or in the early stages of the manuscript.

Reviewer #5: 1. General Assessment

The manuscript titled “Lived experience in mental health research in Ghana and Indonesia: What have we learned?” is a valuable contribution to the field of mental health research, particularly in low- and middle-income countries. It addresses the critical issue of involving lived experience researchers in a methodologically rigorous and ethically responsible manner. The paper aligns with the publication criteria of PLOS Mental Health and offers actionable insights for improving participatory research practices.

2. Technical and Methodological Soundness

The study employs participatory and arts-based methods to engage lived experience researchers in Ghana and Indonesia, effectively addressing an under-researched area in the Global South. The thematic analysis is conducted rigorously, with clear descriptions of how themes were identified and refined collaboratively between peer and academic researchers. The data collected support the conclusions, which are well-drawn and aligned with the study’s objectives. However, the authors should consider elaborating on specific challenges related to thematic saturation and intercultural translation issues to further strengthen the methodological discussion.

3. Statistical Analysis

This is a qualitative study and does not include statistical analysis. The qualitative methods used are appropriate and rigorously applied. The authors’ triangulation of multiple data sources, such as feedback forms, reflections, and discussions, ensures the robustness of the findings.

4. Presentation and Clarity

The manuscript is well-written and presented in standard English. It is clear, concise, and accessible to a broad audience, including readers who may not be familiar with participatory research. The structure—comprising abstract, introduction, methods, findings, discussion, and recommendations—is logical and easy to follow. That said, some sections, particularly the discussion, could benefit from additional specificity about the applicability of the findings beyond the immediate contexts of Ghana and Indonesia.

5. Ethical Considerations

The manuscript demonstrates strong adherence to ethical standards. It includes detailed information on ethical approvals obtained and the informed consent process. The authors also protect participant anonymity by using identifiers rather than names. The manuscript emphasizes the need for equitable remuneration and support for peer researchers, addressing potential concerns about exploitation or tokenism.

6. Strengths

The focus on lived experience researchers in the Global South is novel and addresses a significant gap in the literature.

The participatory methods employed enrich the research process and outputs, offering valuable lessons for future studies.

The recommendations are practical and grounded in the lived experiences of the participants, making them actionable for researchers and policymakers.

7. Areas for Improvement

Capacity Building: The manuscript mentions the importance of training for peer researchers but could provide more detail about the gaps in the training provided and specific ways future studies can address these issues.

Budgetary Considerations: While the authors highlight financial challenges faced by peer researchers, including delayed payments, they could provide more concrete suggestions on how funding mechanisms can better accommodate the needs of peer researchers.

Limitations: The authors acknowledge some limitations, but the discussion could benefit from a deeper exploration of how intersectional factors, such as language barriers and socio-economic status, might have influenced the findings.

8. Recommendations for Publication

I recommend the manuscript for publication with minor revisions to strengthen the discussion and provide additional detail in the methods and recommendations sections.

9. Final Comments

This paper makes an important contribution to the field of mental health research, particularly in ensuring more equitable and inclusive practices in the Global South. The findings are timely and relevant, offering valuable guidance for researchers, practitioners, and policymakers. With the suggested revisions, this paper has the potential to significantly influence future research practices.

Reviewer #6: Overall:

This manuscript addresses an important and under-captured topic. The authors share meaningful insights and practical recommendations for peer research. In particular, the authors provide tangible lessons and recommendations that may be useful to others wanting to integrate participation but not knowing how to do so safely. However, certain aspects of the manuscript could be strengthened to enhance clarity, coherence, and impact. The structure and framing of the results could be improved; reducing repetition and creating clearer links between sections would enhance readability and ensure that the core messages are not lost. The manuscript currently presents findings as though peer researchers were participants rather than co-authors. I encourage the authors to reconsider the framing of the results to reflect the collaborative development of the manuscript. Additionally, expand on how the team defines "peer researcher" providing detail on the extent of involvement, roles, and responsibilities, particularly, as these appear to vary across peer researchers. Overall, I appreciate the author’s thoughtfulness, careful considerations and look forward to seeing the revised manuscript.

Specific Comments

Abstract (pg. 3): The sentence beginning, "Based on these experiences..." could benefit from re-wording for clarity. Consider breaking it into shorter sentences or rephrasing for greater readability.

Introduction (line 30): The terms "Patient and Public Involvement (PPI), community engagement, co-production, participatory research, and peer research" represent distinct approaches and dynamics. Consider briefly clarifying their differences or focusing on the most relevant terms for this study.

Introduction (lines 62-65): While lived experience may enhance empathy and understanding, this assertion would be stronger if supported by evidence – as lived experience is not a necessary precursor for empathy and understanding. Clarify whether there is an assumption of increased "reliability and authenticity", is this based on the assumption or expectation that there will be disclosure of lived experiences? Reflect on the potential risks to emotional safety for peer researchers (and participants) if disclosure becomes an implicit expectation.

Introduction: Consider including mention of the importance of participation and the risks of extractive processes in collaboration in contexts where there is a risk of extractive processes, particularly when Global North institutions benefit from working in Global South contexts.

Methods (line 158): For the WHO Recovery Planning template, as this is part of a self-help toolkit, clarify whether this was for personal reflection (e.g. as a personal safety plan) or if peer researchers were expected to share this information.

Methods (line 178): Provide additional details on the extent of collaboration among peer researchers, academic researchers, and artists in the second project. Specify what is meant by "involved."

Methods (line 198): Expand on how the themes were chosen, detailing decision-making processes and how balance and integration of perspectives (e.g. the integration of Indonesian peer researchers experience into the draft developed by the Ghana team)?

Results: At times the wording shifts between insider and outsider (e.g. line 308 “This confidence was evidenced during participatory thematic analysis of interviews for the first project when peer researchers engaged in lively debates with academic researchers.”) Consider the wording of the results to reflect the collaborative nature of the development of the manuscript (rather than peer researchers as participants).

Results (line 412 “These were not openly discussed by all the peer researchers but were also observed by the academic researchers in their own reflections on the process.”): Expand on whether the shared challenges were collaboratively agreed upon by all researchers. Consider distinguishing challenges experienced by peer researchers from those perceived or experienced by academic researchers.

Results (line 419): Clarify what is meant by “supporting others”. Peer researchers are not peer support workers; explore perceptions or expectations of their roles, such as caregiving or participant support (e.g. line 427 “I needed to be a caregiver”)

Results (line 580 “academic researchers were also experienced psychologists”) Provide additional context - was this part of a pre-planned safety plan? How were dual roles managed and safe boundaries around providing psychological support maintained?

Discussion (line 754): Consider discussing how differences in renumeration might perpetuate power imbalances or be perceived as valuing some opinions over others.

6. PLOS authors have the option to publish the peer review history of their article (what does this mean? ). If published, this will include your full peer review and any attached files.

**Do you want your identity to be public for this peer review?** For information about this choice, including consent withdrawal, please see our Privacy Policy .

Reviewer #1: **Yes: ** Laxmi Pradhan

Reviewer #2: **Yes: ** Noah Hansen

Reviewer #3: **Yes: ** Nancy Clark

Reviewer #4: **Yes: ** Shepard M.M Munyoro

Reviewer #5: **Yes: ** Alberto Gabriel Muanido

Reviewer #6: No

---

## [Editor Report · Decision Letter 1]

Lived experience in mental health research in Ghana and Indonesia: What have we learned?

PMEN-D-24-00484R1

Dear Dr Read,

We are pleased to inform you that your manuscript 'Lived experience in mental health research in Ghana and Indonesia: What have we learned?' has been provisionally accepted for publication in PLOS Mental Health.

Best regards,

Kaaren R Mathias, PhD

Academic Editor

PLOS Mental Health

Thank you for the extensive revisions made in response to reviewers. We are happy to accept this manuscript for publication in its current form. It is an important contribution to expanding the contributions of peer researchers and we thank you for submitting it to this journal.

Yours sincerely,

Kaaren